Genome-wide identification and evolution of WNK kinases in Bambusoideae and transcriptional profiling during abiotic stress in Phyllostachys edulis

Liu RongXiu 1
Vasupalli Naresh 1
Hou Dan 1
Stalin Antony 1 2
Wei Hantian 1
Zhang Huicong 1
Lin Xinchun lxc@zafu.edu.cn 1
1 State Key Laboratory of Subtropical Silviculture, Zhejiang A & F University , Lin’an , Zhejiang , China
2 State Key Laboratory of Subtropical Silviculture, Department of Traditional Chinese Medicine, Zhejiang A & F University , Lin’an , Zhejiang , China
Orlov Yuriy
Electronic publication date: 2022 Jan 13
Publication date: 2022
Volume: 10
Electronic Location ID: e12718
Received 2021 Sep 6; Accepted 2021 Dec 9
Copyright: ©2022 Liu et al.
Copyright year: 2022
Copyright holder: Liu et al.
License: This is an open access article distributed under the terms of the Creative Commons Attribution License, which permits unrestricted use, distribution, reproduction and adaptation in any medium and for any purpose provided that it is properly attributed. For attribution, the original author(s), title, publication source (PeerJ) and either DOI or URL of the article must be cited.
License URL: https://creativecommons.org/licenses/by/4.0/

Keywords: WNK, Moso bamboo, Gene expression, Abiotic stress

Funding: The National Key Research & Development Program of China 2021YFD2200503 The Research Fund for International Young Scientist, the National Natural Science Foundation of China 32150410354 The National Natural Science Foundation of China 31971735 The Natural Science Foundation of Zhejiang Province LZ20C160002 The State Key Laboratory of Subtropical Silviculture ZY20180203 This work was supported by grants from the National Key Research & Development Program of China (2021YFD2200503), the Research Fund for International Young Scientist, the National Natural Science Foundation of China (32150410354), the National Natural Science Foundation of China (31971735), the Natural Science Foundation of Zhejiang Province (LZ20C160002), and the State Key Laboratory of Subtropical Silviculture (ZY20180203). The funders had no role in study design, data collection and analysis, decision to publish, or preparation of the manuscript.

==============================
With-no-lysine (WNK) kinases play vital roles in abiotic stress response, circadian rhythms, and regulation of flowering time in rice, Arabidopsis, and Glycine max. However, there are no previous reports of WNKs in the Bambusoideae, although genome sequences are available for diploid, tetraploid, and hexaploid bamboo species. In the present study, we identified 41 WNK genes in five bamboo species and analysed gene evolution, phylogenetic relationship, physical and chemical properties, cis-elements, and conserved motifs. We predicted the structure of PeWNK proteins of moso bamboo and determined the exposed, buried, structural and functional amino acids. Real-time qPCR analysis revealed that PeWNK5, PeWNK7, PeWNK8, and PeWNK11 genes are involved in circadian rhythms. Analysis of gene expression of different organs at different developmental stages revealed that PeWNK genes are tissue-specific. Analysis of various abiotic stress transcriptome data (drought, salt, SA, and ABA) revealed significant gene expression levels in all PeWNKs except PeWNK11. In particular, PeWNK8 and PeWNK9 were significantly down- and up-regulated, respectively, after abiotic stress treatment. A co-expression network of PeWNK genes also showed that PeWNK2, PeWNK4, PeWNK7, and PeWNK8 were co-expressed with transcriptional regulators related to abiotic stress. In conclusion, our study identified the PeWNKs of moso bamboo involved in circadian rhythms and abiotic stress response. In addition, this study serves as a guide for future functional genomic studies of the WNK genes of the Bambusoideae.

Introduction

Protein kinase is a large superfamily of enzymes known to phosphorylate the threonine, tyrosine, and serine residues of target proteins (Kumar, Raina & Sultan, 2020). They constitute about 4% of the Arabidopsis thaliana proteome and are involved in various functions such as development, cell cycle and signal transduction (Manuka, Karle & Kumar, 2019; Manuka, Saddhe & Kumar, 2015; Wang et al., 2008). A unique subfamily of serine/threonine protein kinases related to the STE20/PAK-like family is called With-no-lysine (WNK) kinases and is found only in multi-cellular organisms (Kumar et al., 2011; Xu et al., 2000). The WNK kinases contain a conserved lysine residue in the subdomain II within the N-terminal domain, which is essential for ATP binding. However, this conserved lysine residue in the active site is absent in the WNK subdomain II (Xu et al., 2000). Moreover, the lysine in subdomain-I is involved in kinase phosphorylation, and it is the characteristic feature of the WNK family (McCormick & Ellison, 2011).

In plants, WNK genes are involved in physiological functions such as maintenance of circadian cycle, root architecture, signal transduction, response to abiotic stress, and flowering time by affecting photoperiod (Kahle et al., 2006; Urano et al., 2015; Urano et al., 2012; Wang et al., 2010). Currently, 11 WNKs are known in A. thaliana and nine WNKs in rice, but only a few genes have been well studied (Manuka, Saddhe & Kumar, 2015). For example, AtWNK1 phosphorylates APRR3 protein, the part of APRR1/TOC1 quintet associated with the clock, to regulate circadian rhythms (Nakamichi et al., 2002). At the same time, the involvement of AtWNK2, AtWNK4, and AtWNK6 in circadian rhythms has also been reported (Nakamichi et al., 2002). Similarly, OsWNK1 shows a rhythmic expression profile under circadian and diurnal conditions and responds to abiotic stress in rice (Kumar et al., 2011).

Furthermore, a knock-out study has demonstrated the importance of AtWNK8 in abiotic stress (Zhang et al., 2013) and overexpression of AtWNK9 increases drought tolerance through the ABA signaling cascade (Xie et al., 2014). In addition, nine WNK(1-9) have been identified in rice that exhibits differential transcriptional regulation for different abiotic stresses such as heat, cold, salt, and drought (Manuka, Saddhe & Kumar, 2015). At the same time, overexpression of OsWNK9 enhances the tolerance to salt, drought, and arsenite in A. thaliana (Manuka et al., 2021; Xu et al., 2000). Similarly, root-specific GmWNK1 in Glycine max regulates root system architecture and stress response via an ABA-dependent signaling pathway (Rodan & Jenny, 2017). At the same time, overexpression of GmWNK1 in A. thaliana showed tolerance towards osmatic and salt stress (Wang et al., 2011). In addition, a total of 114 WNKs were identified from eight fruit tree species. It was predicted that PpWNK.A2 and PpWNK.E3.1 genes might be related to early fruit development, while PpWNK.A1 is likely associated with fruit ripening (Cao et al., 2019).

Bamboos (Bambusoideae) are among the fastest-growing plants globally, and Phyllostachys edulis (moso bamboo) is the most widespread bamboo species in China and has high economic value as edible shoots, timber, and pulp (Choudhury, Sahu & Sharma, 2012). Bamboo can be divided into four monophyletic lineages based on the level of ploidy: diploid herbaceous bamboo, tetraploid temperate and neotropical woody bamboo, and hexaploid paleotropical woody bamboo. Recently, Zhao et al. (2018) reported the chromosome level P. edulis (temperate tetraploid woody bamboo) whole-genome sequence. At the same time, Guo et al. (2019) reported the draft genome sequences of Olyra latifolia and Raddia guianensis (diploid herbaceous bamboo), Guadua angustifolia (tetraploid neotropical woody bamboo) and Bonia amplexicaulis (hexaploid paleotropical woody bamboo). Due to climate change, naturally growing bamboo species were subjected to different kinds of abiotic stress. Recently, Liu et al. (2019) reported that the P. edulis yield and the quality of winter shoots were severely affected by abiotic stress conditions. Therefore, studying the genes involved in abiotic stress in bamboo species is helpful to develop better adapted genetically modified bamboo plants to the changing environment. The availability of the chromosome level genome of P. edulis, draft genome sequences of other bamboo species, and various transcriptomic data from tissues provide the opportunity for genome-wide analysis WNK genes (Guo et al., 2019; Zhao et al., 2018). In this study, we identified 41 WNK genes belonging to the five bamboo species. Then, we analysed the physicochemical properties, protein structure, and evolution of the WNKs of the Bambusoideae. We also analysed the expression of PeWNKs genes in different tissues, the response to abiotic stress, and the co-expression network. The present study results provide a basis for the functional analysis of WNK genes in P. edulis.

Materials & Methods

Plant materials

P. edulis seeds used for transcriptomic data were collected in Linchuan County, Guangxi Zhuang Autonomous Region, China. For qPCR analysis, P. edulis leaves were collected from the Cuizhu Garden of Zhejiang Agriculture and Forestry University. Samples were collected every four hours, from 6 AM on April 25, 2021, to 48 h.

Identification of WNK genes from P. edulis genome databases

The WNK genes of A. thaliana and rice were downloaded from the Phytozome (https://phytozome.jgi.doe.gov/pz/portal.html). We used the genome database of P. edulis and transcriptomic data (Zhao et al., 2018) to identify WNK family genes through the local BLAST analysis. At the same time, other WNK genes of the Bambusoideae were isolated from draft sequences of the herbaceous diploid bamboo species O. latifolia and R. guianensis and the tetraploid and hexaploid woody species G. angustifolia and B. amplexicaulis (Guo et al., 2019). The candidate genes obtained were verified against the NCBI database (https://www.ncbi.nlm.nih.gov/). The amino acid sequences of WNK genes were aligned to confirm conserved regions. The sequences without a complete reading frame and conserved domain were removed.

Physicochemical properties, phylogenetic tree and motif analysis of WNK genes

The amino acid number, molecular weight, and isoelectric point of PeWNK proteins were calculated using the online software ExPASy (https://www.ExPASy.org/). The phylogenetic tree was constructed using the maximum-likelihood method with MEGA-X (Kumar et al., 2018). The conserved domains of plant species A. thaliana, Glycine max, Oryza sativa, Zea mays, P. edulis, R. guianensis, O. latifolia, G. angustifolia and B. amplexicaulis were used to construct the phylogenetic tree. A bootstrap value of 1,000 replicates was calculated to evaluate the statistical significance of clade level relationships. Subsequently, the phylogenetic tree for WNKs was imported into the ITOL server (http://itol.embl.de/). The conserved motifs were identified using the MEME server and visualized in TBtools (Chen et al., 2020).

Protein secondary and tertiary structure of PeWNK genes

The secondary structures of the WNK proteins of P. edulis were predicted through the online website SOPMA (https://npsa-prabi.ibcp.fr/NPSA/npsa_sopma.html) with the default parameters of four conformational states (helix, sheet, turn, coil) and similarity threshold eight. The tertiary structures of the WNK proteins of P. edulis were predicted using the Modeller tool with the help of the Consurf server (Berezin et al., 2004). The models of the proteins were built based on the ’ConSeq’ mode and the given selected parameters were used to build the multiple sequence alignments. The homologs were taken from the UniProt database and CS-BLAST was used as the algorithm for homolog search (CSI-BLAST E-value: 0.0001; No. of CSI-BLAST Iterations: 3; maximal percentage ID between sequences: 95; minimal percentage ID for homologs: 35; 150 sequences querying the list of homologs for retrieval. For phylogenetic tree analysis, Neighbor-Joining with ML distance algorithm was used. Bayesian computational calculation and best-fit model of substitution for proteins were used to calculate the conservation scores.

Analysis of Cis-acting element

We retrieved the upstream sequence region (2 Kb) of the WNK genes from the genome database to analyse the cis-acting elements. The retrieved sequences were analysed using the PlantCARE program (http://bioinformatics.psb.ugent.be/webtools/plantcare/html/) to identify the putative cis-acting elements. The cis-elements related to ABA, GA, SA and circadian rhythms were visualized through TBtools (Chen et al., 2020).

The PeWNK gene expression in different tissues

Transcriptome data of 26 different tissues of P. edulis were obtained from the NCBI Short Read Archive database (SRX2408703) (Zhao et al., 2018) and used for tissue expression studies. The FPKM values of the WNK genes of P. edulis were used to develop a heat map using TBtools (Chen et al., 2020).

Expression analysis of PeWNK genes in response to abiotic stress

Thirty-day old equal height P. edulis seedlings were used for abiotic stress treatment. Seedlings were treated with 25% polyethylene glycol (PEG), 200 µM Abscisic acid (ABA), 1 mM salicylic acid (SA) (unpublished) and 200 mM sodium chloride (NaCl) (Yang et al., 2010) nutrient solution for 3 h and 24 h, respectively. Total RNA was isolated from young leaves and RNAseq data were generated on the Illumina platform (pair-end reads) in three biological and technical replicates (GSE169067). The adapter sequences and low-quality reads were removed and the high-quality reads were mapped to the reference genome sequence using the Hisat2 tool. FPKM values of the RNAseq data were developed and used to generate graphs.

Real-time qPCR analysis

Total RNA from leaf samples was isolated using Trizol reagent. According to the manufacturer’s instructions, cDNA was synthesised using the PrimeScript RT reagent kit with gDNA Eraser (TaKaRa, Shiga, Japan). The 2XNovoStart SYBR qPCR SuperMix Plus (novoprotein, Suzhou, China) was used for qRT-PCR amplification in a real-time PCR instrument (BioRad, USA). The qPCR reaction conditions are as follows: initial denaturation 95 °C for 5 min, followed by 40 cycles of 30 s at 94 °C, 30 s at 60 °C, and 30 s at 72 °C. A melting curve was included from 65 to 95 °C to check amplification specificity. The 2−ΔΔCt method was used to determine the relative expression levels. In addition, NTB was used as a reference gene in P. edulis according to previous studies (Zhao et al., 2019). The qPCR primers for the PeWNK genes used for gene expression analysis are listed in Table S1.

Co-expression analysis of PeWNK genes

We submitted the PeWNK genes to the BambooNET (http://bioinformatics.cau.edu.cn/bamboo/index.html) and acquired the co-expression network data.

Results

Identification of the Bambusoideae WNK genes

The genome database of P. edulis and the draft genomes of R. guianensis, O. latifolia, G. angustifolia, and B. amplexicaulis were used to find the WNK candidate genes in the Bambusoideae. In addition, the WNK genes of A. thaliana and rice were downloaded from Phytozome and used as reference genes to identify the WNK genes of the Bambusoideae through the local BLASTP. The sequences containing the serine/threonine-protein kinase domain are referred to as Bambusoideae WNK genes (PeWNKs, RguWNKs, OlaWNKs, GanWNKs, and BamWNKs) (File S1). A total of 11 WNK genes of P. edulis (PeWNK1-11) and 30 WNK genes of the other four bamboo species were identified. The WNK proteins of the Bambusoideae range from 257 to 1905 amino acids, of which RguWNK1 is the smallest and PeWNK8 is the largest. At the same time, the molecular weight is 29047.42 and 157,857.24, respectively. Moreover, the isoelectric point and instability index are 4.56 to 6.74 and 29 to 59.63, respectively. In addition, the aliphatic index and the grand average of hydropathicity are 19.42 to 95.95 and −0.647 to 0.977, respectively (Table S2). Furthermore, we identified that PeWNK genes were located on nine scaffolds, with scaffolds 4 and 10 containing two genes, whereas the remaining scaffolds contained only one gene (Fig. S1).

Figure 1 The phylogenetic tree of WNK genes from dicot and monocot plants.

The phylogenetic tree was constructed using WNK sequences of Arabidopsis thaliana (At), Glycine max (Gm), Oryza sativa (Os), Zea mays (Zm), P. edulis (Pe), O. latifolia (Ola), R. guianensis (Rgu), G. angustifolia (Gan) and B. amplexicaulis (Bam). The bootstrap support values were mentioned as the numbers on the branches. Clade I, II and III are indicated in the blue, violet and pink colours, respectively. The dicot plants WNK genes were indicated in the grey colour boxes and the diploid bamboo species are indicated in the yellow colour boxes. The conserved motifs (1–10) are mentioned in different colour boxes.

Evolution of WNK gene family

To understand the evolution of WNK genes, a total of 78 WNK genes (AtWNKs, GmWNKs, OsWNKs, ZmWNKs, PeWNKs, BamWNKs, GanWNKs, OlaWNK, and RguWNKs) were used to construct the phylogenetic tree (File S1). The highly aligned peptide sequences were used to generate a phylogenetic tree using the maximum likelihood method with 1,000 bootstrap replicates (Fig. 1). The WNKs were mainly divided into three clades, namely clade I, II and III. In addition, clade III was divided into clades IIIA and IIIB and clade III has more genes than clades I and II. Additionally, all clades were supported by high bootstrap values. Based on the topological structure, the evolution of WNK genes was clearly divided between monocots and dicots in the phylogenetic tree. In clades, I, II, IIIA, and IIIB, monocot and dicot WNK genes were divided into two sub-branches with higher bootstrap values. These results suggest that WNKs were present before the divergence of monocot and dicot plants. Moreover, the OlaWNKs and RguWNKs of herbaceous bamboo were also divided into sub-branches compared with the other woody bamboo species. This suggests that WNKs evolved separately after polyploidisation in the Bambusoideae (Fig. 1).

The evolution of plant species is driven by polyploidisation, including in the Bambusoideae (Ramakrishnan et al., 2020). In the phylogenetic tree, the WNKs of diploid and polypoid bamboo species were also separated by sub-braches. Moreover, the copy number of WNKs was increased in the tetraploid P. edulis and hexaploid B. amplexicaulis compared to the diploid bamboo species O. latifolia and R. guianensis. In contrast, the copy number of WNKs is surprisingly lower in the tetraploid G. angustifolia than in the diploid bamboo species. Furthermore, we analysed the evolution of specific domains in WNKs between dicot and monocot plants (Fig. 1). Using the MEME server, we identified ten conserved motifs in the WNKs proteins. With few exceptions, most WNKs in all three clades contain all ten domains in the same serial order. RguWNKI in clade II and GmWNK4 in clade III contain the least number of six domains. BamWNK3 in clade I, on the other hand, has 17 domains, with domains 1, 2, 3, 5, 7, 8, and 10 were duplicated. In addition, the starting domain nine is absent in most of the monocot groups of clade I. In contrast, in clade II, the last two domains 7 and 10 are missing in half of the bamboo WNKs.

Cis-acting elements responsive to abiotic stress and circadian rhythm

Cis-acting elements affect genes involved in the stress response. Hence, studying the cis-acting elements in the promoter region helps to understand the role of WNK genes in the stress response. Therefore, we analysed the putative cis-elements in the 2 kb region upstream of the translational start site of WNK genes in both monocot (OsWNKs, PeWNKs, BamWNKs, GanWNKs, OlaWNK, and RguWNKs) and dicot (AtWNKs and GmWNKs) plants (File S2). Among them, we focused on exploring the ABA, GA, SA and circadian rhythm responsive elements, and there are several cis-elements associated with them in WNKs. For example, ABA-responsive elements (ABREs) are present in 61 genes, including all 11 PeWNK genes (File S3). We, therefore, hypothesise that ABA stress responses regulate most WNK genes. Moreover, the GA responsive GARE-motif, P-box, and TATA-box elements are present in the promoter regions of 21, 18, and 71 WNK genes, respectively. Similarly, SA responsive element TCA is present in the promoter region of 24 WNK genes. Interestingly, 12 WNK genes also have cis-acting elements associated with circadian control (Fig. 2). Further, GC-motif and SP1 are present in some of the monocot WNK and GmWNK genes but absent in Arabidopsis (Fig. S2).

Figure 2 The conserved cis-elements analysis of WNK genes in the promoter regions of Bambusoideae and other monocot and dicot plants, related to stress response (P-box, ABRE, GARE-motif, TCA and circadian).

Prediction of the protein structure of P. edulis WNK

The secondary structure of the protein plays an essential role in constructing the tertiary structure of the protein and its normal function. It mainly consists of hydrogen bonds and the primary forms include α-helix, β-turn, random curling, etc. The secondary structure of 11 P. edulis WNK proteins was predicted using the online website SOPMA (Table 1). It can be seen that the WNK protein of P. edulis has a relatively similar protein secondary structure. Modeller 9.19 software was used to predict the tertiary structure of 11 identified WNK proteins of P. edulis (Ashkenazy et al., 2016). Furthermore, we compared and analyzed the domains and motifs of PeWNK1 with human_WNK3, GmWNK1, OsWNK9 and AtWNK1 (Fig. 3). The PeWNK1 sequence was similar to all other previously published WNKs genes (Manuka, Saddhe & Kumar, 2018). PeWNK1 has an N-terminal protein kinase domain divided into 12 subdomains. In addition, an activation loop (A-loop), an autoinhibitory conserved domain-containing FXF motif, the ’IIHRDLKCDNIFI’ motif in subdomain VIb and the ‘GTPEFMAPE’ motif in subdomain VIII were conserved (Fig. 3). Besides, we compared the eleven WNK genes from P. edulis in which all these A-loops and motifs were conserved (Fig. S3), and we also detected that these A-loops and motifs were conserved in all monocot and dicot plants used in this study (File S1). Moreover, we also analysed the phosphorylation sites of the PeWNK proteins and identified that, except for PeWNK9, all other WNKs contained the phosphorylation sites (Fig. 4 and Fig. S3).

Table 1 Protein secondary structure of WNK protein in P. edulis.

Protein	ID of gene	Alpha helix	Beta turn	Extended strand	Random coil	
PeWNK1	PH02Gene37861.t1	36.12%	3.74%	8.92%	51.22%	
PeWNK2	PH02Gene17877.t1	37.07%	5.75%	11.65%	45.54%	
PeWNK3	PH02Gene03314.t1	41.44%	3.75%	9.30%	45.51%	
PeWNK4	PH02Gene01510.t1	38.27%	3.91%	10.10%	47.72%	
PeWNK5	PH02Gene07448.t1	37.37%	5.30%	11.65%	45.69%	
PeWNK6	PH02Gene25768.t1	38.12%	4.06%	8.59%	49.22%	
PeWNK7	PH02Gene38251.t1	37.69%	3.90%	9.85%	48.56%	
PeWNK8	PH02Gene03413.t1	35.49%	4.57%	11.04%	48.90%	
PeWNK9	PH02Gene20314.t1	42.60%	5.47%	10.95%	40.98%	
PeWNK10	PH02Gene23702.t1	40.21%	3.96%	10.83%	45.00%	
PeWNK11	PH02Gene11468.t1	36.32%	6.21%	16.32%	41.15%	

Figure 3 Multiple sequence alignment between PeWNK1, HUMAN_WNK1, GnWNK1, OsWNK9, and AtWNK1 protein sequences.

Conserved domains, motif and secondary structural arrangements were highlighted. The phosphorylation sites were mentioned in the blue background.

Figure 4 Conserved domain sequence analysis of WNK in the Bambusoideae (PeWNK1) protein predicted by Consurf server.

Predicted homology model of PeWNK1 using modeler; highly conserved WNK kinase domain and autoinhibitory domain were highlighted.

Furthermore, all PeWNK protein sequences were compared with known WNK proteins in the Uniprot database using ConSurf domain analysis (Ashkenazy et al., 2010; Celniker et al., 2013). Based on the phylogenetic relationship between the homologous sequences of WNK, the conserved regions of amino acids were identified. For instance, the conserved domain region of PeWNK1 is shown in the colour magenta (Fig. 4). The remaining conserved domains of the ten PeWNK proteins are listed in a File S4. As mentioned in Fig. 3, most of the amino acids in the activation loop, autoinhibitory domain (FPF), and kinase domain are located in the conserved region. We also identified the exposed, buried, functional and structural (e, b, f, s) residues/amino acids in the PeWNKs. All functional residues are the exposed residues, while all structural residues are buried (Fig. 4).

PeWNK genes response to circadian rhythms

WNK genes have been previously reported to be involved in circadian rhythms (Kumar et al., 2011; Nakamichi et al., 2002). Therefore, we collected leaf samples of P. edulis every four hours starting from 6 AM up to 48 h and conducted qPCR experiments to identify the PeWNK genes of P. edulis involved in circadian rhythms. The results showed that among the 11 PeWNK genes of P. edulis, PeWNK5, PeWNK7, PeWNK8, and PeWNK11 follow circadian rhythms (Fig. 5). The PeWNK7, PeWNK8, and PeWNK11 genes show a clear circadian expression pattern in the morning, with a peak forming every 0 and 4 h (6 and 10 AM). In contrast, the expression pattern of PeWNK5 follows a 12 h cycle. After 0 h in the morning, the expression drops to a very low level at 4 h and increases again at 8 and 12 h (2 and 6 PM) (Fig. 5).

Figure 5 Expression analysis of PeWNK genes for the circadian cycle.

qRT-PCR analysis of PeWNK5, PeWNK7, PeWNK8 and PeWNK11 genes normalized with NTB. Moso bamboo cDNA leaf samples 0–48 h. The error bar indicates the standard deviation (n = 3).

Expression profile of PeWNK genes in different tissues

To elucidate the expression profiles of PeWNKs in different tissues, we developed a heatmap using transcriptomic data from 26 different tissues at different developmental stages, as mentioned by Zhao et al. (2018). The heatmap indicates that some PeWNK genes have high expression in specific tissues. For example, the expression patterns of PeWNK10 were very high in the middle and lower portion of the 3 m shoot, while the expression in the other tissues was comparatively low. In addition, PeWNK7 was expressed in the rhizome, whereas PeWNK6 and PeWNK1 were mainly expressed in the leaf. Interestingly, the expression of PeWNK genes was relatively low in the rhizome bud (budR), lower bud, and top 3m shoot (Fig. 6).

Response of PeWNK genes under abiotic stress treatments

We analysed the transcriptomic data to investigate further the characteristics of PeWNK gene expression in P. edulis seedlings under drought, salt, SA and ABA treatments. The analyses showed that PeWNK genes responded differently at 3 h and 24 h after exposure to drought, salt and hormone stress. In this study, the genes with two-fold differences were considered to be differentially expressed compared with the control (Wang et al., 2020). Among all PeWNK genes, the expression of PeWNK9, in particular, was significantly up-regulated after abiotic stress treatments (Fig. 7). Under PEG, NaCl and ABA treatment, the relative expression of PeWNK9 was up-regulated 146, 117, and 307 times respectively, after 24 h compared with the control. Similarly, the relative expression of PeWNK4 was up-regulated by 2.2–8.2 times of control after 3 h in all treatments. Further, the relative expression of PeWNK7 and PeWNK8 was significantly downregulated after 3 h in all treatments. After 24 h of treatment with SA, the expression of PeWNK7 was up-regulated to 2.6 times and the expression of PeWNK8 was downregulated to 0.45 times of control (Fig. 7).

Figure 6 Expression of PeWNK genes in 26 different tissues and stages of bamboo growth.

The log2 expression values represent each colour box and the colour scale is present on the upper right side.

Figure 7 Expression analysis of PeWNK genes in response to Polyethylene glycol (PEG), Sodium chloride (NaCl), Abscisic acid (ABA) and Salicylic acid (SA).

The FPKM values of transcriptomic data (Moso bamboo seedlings treated with PEG (25%), NaCl (200 mM), ABA (1uM), SA (1 mM) for 3 h and 24 h) are used to develop graphs. The error bar indicates the standard deviation (n = 3).

After 3 h of treatment with SA, the relative expression of PeWNK1 was downregulated to 0.23 times of control. Similarly, the expression of PeWNK2 was significantly downregulated after 3 h of PEG, SA, ABA and 24 h of NaCl treatment. Likewise, the expression of PeWNK5 and PeWNK6 was downregulated to 0.4 times after 3 h of SA treatment. While the expression of PeWNK6 was up-regulated to 2.5 times of control after 3 h of ABA treatment. The expression of PeWNK10 was significantly up-regulated after both NaCl treatment and 24 h SA treatment. At the same time, expression was downregulated to 0.4 times of control after 3 h treatment with PEG and 24 h treatment with ABA. The expression levels of PeWNK11 are too low for analysis (Fig. 7).

Co-expression analysis of PeWNK genes

A co-expression network has been successfully applied to identify the transcription factors or regulators in many plant species (Bishop et al., 2020; Gao et al., 2020; Yang et al., 2017). To determine the regulators of PeWNK genes, we used the BambooNET database. The 11 PeWNK genes were searched for transcriptional regulators in the BambooNET database. PeWNK8 (PH02Gene03413.t1) is co-expressed with 17 genes, including GRAS family transcription factor and F-box protein 2 (Fig. 8). Interestingly, both genes have been reported to be associated with abiotic stress. Similarly, both PeWNK2 (PH02Gene17877) and PeWNK4 (PH02Gene23702) were co-expressed with an F-box family protein (PH02Gene00258). Furthermore, PeWNK7 (PH02Gene03314) is co-expressed with the PEBP (phosphatidylethanolamine-binding protein) family protein and the myb domain protein 48 (File S5). These two proteins are involved in the suppression of flowering and circadian rhythms, respectively.

Figure 8 Co-expression network of PeWNK8 (PH02Gene03413.t1), PeWNK2 (PH02Gene17877), PeWNK4 (PH02Gene23702), PeWNK7 (PH02Gene03314).

The boxes indicate the genes involved in abiotic stress response.

Discussion

Bamboo is one of the fastest-growing perennial plants and has the longest vegetative stage before flowering (Liu et al., 2019; Ramakrishnan et al., 2020). However, the mechanisms involved in abiotic stress during bamboo growth are poorly understood. WNK genes, which belong to the serine/threonine protein kinases of the STE20/PAK-like subfamily (Manuka, Saddhe & Kumar, 2015) play an essential role in regulating plant salt tolerance and osmotic stress by coordinating ion channels and signal transduction during the transportation process (Kahle et al., 2006; Wang et al., 2010). In addition, WNK genes are also involved in circadian rhythms (Nakamichi et al., 2002). To date, WNK genes have been identified in Arabidopsis, rice, soya bean, and fruit trees (Cao et al., 2019; Kumar et al., 2011; Wang et al., 2008). However, the identity and function of WNKs in bamboo, including P. edulis, have not yet been identified. In this study, we identified WNK genes in diploid and polyploid bamboo species and investigated the evolution of WNKs between monocot and dicot plants. Further, we identified the protein structure, response to abiotic stress, tissue-specific expression, and co-expression analysis of PeWNK genes in P. edulis.

We identified a total of 41 WNK genes from the available bamboo genome database and investigated their gene evolution, physical and chemical properties, and conserved motifs. The putative amino acid lengths of WNKs from Rice, G. max, and Populus trichocarpa range from 328–705, 480–738 and 297–739 amino acids, respectively (Manuka, Saddhe & Kumar, 2015; Wang et al., 2010). At the same time, human WNK1 has a length of 2,382 amino acids (VerõÂssimo & Jordan, 2001). In our study, the length of the amino acids of WNK of diploid bamboo is 257–702, that of tetraploid bamboo is 285–1905, and that of hexaploid bamboo is 290–739. These results suggest that the amino acid lengths of diploid, hexaploid and tetraploid GanWNKs are similar to those of rice and G. max. Interestingly, the amino acid length of the four PeWNK genes in P. edulis ranges from 1771–1905, which is almost the size of human WNKs and three times longer than OsWNK.

PeWNKs have the N-terminal protein kinase domain, which has the altered lycine residue in the Gly-X-Gly-X-X-Lys-X-Val motif of subdomain I instead of Gly-X-Gly-X-X-Gly-X-Val. In addition, the WNK genes of higher plants were divided into three clades. These results are consistent with previous findings in plants and animals (Manuka, Saddhe & Kumar, 2015; Xu et al., 2000). Moreover, the distribution of conserved motifs was similar among WNK proteins in the same clade. These results and phylogenetic analysis support the reliability of clade classification and the similar functions of proteins in the same clade. Moreover, the number of genes in the gene families increased with the duplication events and polyploidization (De Grassi, Lanave & Saccone, 2008; Li et al., 2020). The copy number of WNKs was increased in the tetraploid P. edulis and hexaploid B. amplexicaulis compared to the diploid bamboo species O. latifolia and R. guianensis. In contrast, the copy number of WNKs is lower in the tetraploid G. angustifolia than in the diploid bamboo species. These results might be due to low coverage, poor sequencing, and incomplete genome database.

Tissue-specific expression analysis of OsWNK genes in rice revealed that most OsWNK genes are more highly expressed in roots than in other tissues, indicating the role of OsWNKs in root formation and architecture (Manuka, Saddhe & Kumar, 2015). In Arabidopsis, AtWNK8 is mainly expressed in the hypocotyl, primary root, and pistil (Zhang et al., 2013). At the same time, all other AtWNK genes (except AtWNK6) are expressed in different tissues and organs at different developmental stages (Wang et al., 2008). In the fruit tree Prunus persica, gene expression analysis revealed that PpWNK.A1 is probably involved in fruit ripening, while PpWNK.A2 and PpWNK.E3.1 are associated with early fruit development (Cao et al., 2019). In contrast to rice OsWNKs, tissue-specific expression analysis of PeWNK genes in our study shows that most PeWNK genes are expressed only in a particular tissue at a specific plant height, indicating diverse roles in different developmental stages of the tissues.

Various abiotic stress conditions severely affect P. edulis yield and the quality of winter shoots (Liu et al., 2019). Protein kinases in plants play a crucial role in stress-induced signal transduction pathways (Kumar et al., 2013). Our results showed that all PeWNK genes responded to abiotic stress, except PeWNK11. A T-DNA knock-out mutant study showed that AtWNK8 was induced after salt and sorbitol stress, and disruption of AtWNK8 enhances tolerance to NaCl and osmotic stress (Zhang et al., 2013). Moreover, overexpression of OsWNK9 increases tolerance to salt, drought, and arsenite in transgenic Arabidopsis plants (Manuka, Karle & Kumar, 2019; Manuka et al., 2021). Phylogenetic analysis of the gene family shows that AtWNK8 and OsWNK9 are closely related to PeWNK7, PeWNK8, and PeWNK9. Our study also provided evidence that the expression of PeWNK9 was significantly increased after all abiotic stress treatments. In contrast, the expression of PeWNK8 significantly decreased considerably after 3 h of PEG, NaCl and SA treatments. Similarly, the OsWNK1 gene was up-regulated after drought and cold stress and downregulated after salt stress (Kumar et al., 2011). Both PeWNK1 and PeWNK2 were similar to OsWNK1 and both were significantly downregulated after all abiotic stresses studied. These results suggest that these proteins have similar functions and are predominantly involved in abiotic stress response.

In addition, our co-expression network analysis also revealed the relationship between abiotic stress genes and PeWNK genes. In this study, PeWNK8 was found to be co-expressed with transcription factor GRAS and F-box protein 2. The transcription factor OsGRAS23 from rice is involved in drought stress response, and the transcription factor GRAS from Vitis amurensis induces abiotic stress tolerance in Arabidopsis (Xu & Zhang, 2015; Yuan et al., 2016). Similarly, an F-box protein MAX2 regulates drought tolerance in Arabidopsis (Bu et al., 2014). Interestingly, PeWNK8 was downregulated after PEG, NaCl and SA treatments, indicating its involvement in the abiotic stress response.

Conclusions

In the present study, we identified 41 WNK genes in five Bambusoideae species and analyzed the conserved motifs, domains, cis-acting elements, and tissue-specific expression studies. The qRT-PCR analysis revealed that PeWNK5, PeWNK7, PeWNK8, and PeWNK11 are involved in circadian rhythms. Transcriptome analysis of different abiotic stresses and co-expression analysis also revealed that PeWNK8 and PeWNK9 are involved in abiotic stress response. Thus, these genes can be used as good candidates for the production of genetically modified and economically important bamboo plants.

Supplemental Information

Supplemental Information 1 List of WNK genes used in this study and their conserved motifs

Click here for additional data file.

Supplemental Information 2 List of cis-elements for WNK genes used in this study

Click here for additional data file.

Supplemental Information 3 List of WNK genes contains the cis-elements (ABRE, GARE-motif, P-box, TATA-box, TCA-elements, Circadian, GC-motif and Sp1)

Click here for additional data file.

Supplemental Information 4 Conserved domains of PeWNK genes

Click here for additional data file.

Supplemental Information 5 List of genes co-expressed with PeWNK genes

Click here for additional data file.

Supplemental Information 6 Details of qPCR primers

Click here for additional data file.

Supplemental Information 7 The physical and chemical properties of WNK protein in Bambusoideae

Click here for additional data file.

Supplemental Information 8 PeWNK genes location on the scaffolds

Click here for additional data file.

Supplemental Information 9 The conserved GC-motif and Sp1 cis-elements distribution in the promoters of WNK genes

Click here for additional data file.

Supplemental Information 10 Multiple sequence alignment between PeWNK protein sequences

Conserved domains, motif and secondary structural arrangements were highlighted. The phosphorylation sites were mentioned in the yellow background.

Click here for additional data file.

Additional Information and Declarations

Competing Interests

Author Contributions

Data Availability

The authors declare there are no competing interests.

RongXiu Liu and Antony Stalin performed the experiments, analyzed the data, prepared figures and/or tables, authored or reviewed drafts of the paper, and approved the final draft.

Naresh Vasupalli conceived and designed the experiments, performed the experiments, analyzed the data, prepared figures and/or tables, authored or reviewed drafts of the paper, and approved the final draft.

Dan Hou and Xinchun Lin conceived and designed the experiments, authored or reviewed drafts of the paper, and approved the final draft.

Hantian Wei and Huicong Zhang analyzed the data, authored or reviewed drafts of the paper, and approved the final draft.

The following information was supplied regarding data availability:

The raw data is available in the Supplementary File. The raw reads are available at GenBank: GSE169067.

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
