# Peer review of "Genome-wide identification and evolution of WNK kinases in Bambusoideae and transcriptional profiling during abiotic stress in Phyllostachys edulis"

_PeerJ, doi:10.7717/peerj.12718_

## Round 0.1 · original submission · Major Revisions

We have received three reviews with critical comments suggesting major revision. I believe the authors should update the manuscript based on these remarks. An additional recommendation is to check the English presentation.

Reviewer 1 ·

Basic reporting

The manuscript ' Genome-wide identification and evolution of WNK kinases in Bambusoideae and transcriptional profiling during abiotic stress in Moso bamboo' is an interesting a well-designed study that could shed light on the role of WNK kinases in Bambusoideae, and in WNK kinases evolution in the plant kingdom.

Experimental design

As far as I know, the manuscript is relevant and up to date, and despite being an article similar to others already published, addressing the same target in other plant species, the results are unprecedented for moso bamboo. However, I think that this study would need some improvement.

Validity of the findings

-First of all, you ignored the role of WNK kinases in circadian rhythms, which was mentioned in the introduction. I think you should consider circadian rhythms when designing the abiotic stress responsing experiment. The treatment time should be at least up to 48 hours.
-I am a little confused about the sentence in lines 323-324 “we identified 41 WNK genes in five species of Bambusoideae, 324 including 11 PeWNK genes in Moso bamboo.” Does the sequences of 11 PeWNK genes in Moso bamboo different from those in the other four species? It is better to compare the sequences of WNK genes, which one is the same and which one is different.
-the reference should be uniform. I find some of the Journal names are the abbreviation some are not.
-the word “isolate” is not proper in lines 86, 96, 145…….
- it is preferred to have a table or a figure to introduce the name of WNK genes in five species. Such as the species name of BamWNKs, GanWNKs, OlaWNK and RguWNKs.
-why was only PeWNK1 analyzed in Figure4 and 5?
-line 229-230 “In this study, the genes with the two-fold difference compared with the control were considered differentially expressed.” should have reference.

Additional comments

-I would also suggest thorough English proofreading since I found several misspellings and confusion regarding the use of adverbs, nouns, and adjectives. I am not a native speaker and I know how hard could be to write in English, and I am not requiring perfect English but the most correct possible.

Below you can find some of the errors in your MS

1. LINE 13 change “plays” to “play”
2. Line 18 “cis-elements” the cis should be italic
3. Line 35 “Arabidopsis thaliana”, line 75 “Phyllostachys edulis”, line 81-82, line94-95,line 97-99 ………… should be italic
4. Line 78-80 the reference format should be uniform.
5. Because you don’t have any wet experiments in your MS. I think more details should be added in the Method parts of lines 129-135. It is hard to understand the Abiotic Stress process for readers
6. Line 124: you should say “The PeWNK gene expression in different tissues”
7. Space should be added between the number and the unit. Line 131, 306,……
8. The general rule is to capitalize the first letter of the first word in a title or heading, the first letter of all other words in a title or heading except conjunctions, articles, prepositions of fewer than four letters, and the “to” in infinitives.

Reviewer 2 ·

Basic reporting

1. Title: I think that “Moso bamboo” should be “moso bamboo”?
2. Line 132: 1uM should be 1μM.
3. Line 207: I think GnWNK1 should be GmWNK1.
4. Latin names of species should be italicized.
5. English editing is needed. Please check the spelling and grammar through the whole manuscript carefully. I suggest that the paper should be polished by native English speakers.

Experimental design

1. Line 129: Please add the source background of plant materials. What are the bases for selecting reagent concentration and treatment time in this section?
2. Figures: In figure 3, Why did you choose HUMAN_WNK1 for alignment? I think the figures should be better reorganized in the revised manuscript.
3. In this manuscript, 41 WNK genes were identified from five bamboo species, but only the WNK genes in moso bamboo were studied in detail. Why? Is the stress resistance of moso bamboo stronger than the other four bamboo species?

Validity of the findings

no comment.

·

Basic reporting

The paper entitled “Genome-wide identification and evolution of WNK kinases in Bambusoideae and transcriptional profiling during abiotic stress in Moso bamboo”by Liu identified 41 WNK genes in five bamboo species and analyzed the gene evolution, phylogenetic relationship, physical, cis-elements, conserved motifs, various abiotic stresses, PeWNK genes expression levels. In this regard, the new reported results would be very useful in future functional genomic studies of Bambusoideae WNK genes.The work is suitable for the journal and the results are interesting.
However, as written, the paper needs major revision. I have the following suggestions that the authors may want to consider when revising the manuscript:
Abstract is not concise, and the main idea of the paper is not clearly visible: the long introduction about only the “WNK” of the object of the study; the results are not summarized in appropriate way and the conclusion is too ambitious.

Experimental design

First mandatory action is the through revision of the English grammar. There are many instances in which the English is very poor and compromises understanding what the authors want to say due to the amount of formatting errors that exist (lack of spaces, names without italics, etc. etc.) . For instance, the sentence of the abstract”While overexpression of AtWNK9 increases drought tolerance through the ABA signaling cascade, nine WNK(1-9) were identified in rice and showed differential transcriptional regulation for various abiotic stresses such as heat, cold, salt, and drought”. I strongly recommend the authors to find a native English speaker to proofread the manuscript.
Materials & Method section. The methodology part needs to be reframed and the authors should provide the bioinformatics part along with the parameters used for the same. Also, check basic rules for botanical publishing. The genus and species names should be written with italics in the text. For instance, “At the same time, other WNK genes of the Bambusoideae were isolated from draft sequences of the herbaceous diploid bamboo species O. latifolia and R. guianensis, and the tetraploid and hexaploid woody species G. angustifolia and B. amplexicaulis [24].” Also,in the part “Expression Analysis of PeWNK Genes in Response to Abiotic Stress”, what is the basis for selection“ 25% Polyethylene glycol (PEG), 200mM Sodium chloride (NaCl), 1uM Abscisic acid (ABA) and 1mM Salicylic acid (SA) nutrient solution” ? and also for just only “for 3 h and 24 h, respectively”?

Validity of the findings

Results section.Figures in the Results section should be rechecked (e.g. lack of the legend in Fig. 1, etc. etc. ) , what’t the difference for the evolution of specific domains in WNKs between the diploid and polyploid bamboo species? All the N-terminal domain and C-terminal domain are the same, respectively? Are there some conseverd palnt WNK motif in C-terminal domain?
Discussion should be more focused on the interpretation of the results. Here, the results were presented again without a clear connection with the previous studies and the importance of the present study.
The conclusions section, as written, is only a plain summary of the results. Instead, it should additionally state the significance of results and what are your thoughts/ perspectives for the field given the results.Reference list should be rechecked and corrected according to the journal style.

Additional comments

There are several ways how this paper can be improved.
For example, you can add the phosphorylation sites of protein kinase PeWNK .And also, the chromosomal level of Moso bamboo (temperate tetraploid woody bamboo) as a whole-genome sequence had been reported. You can add the map of the distribution of PeWNK gene on chromosomes if you can.
Did you calculate any conservative sequence analysis in all plant WNKs family or the selected five Bambusoideae species? Are there any amino acids that are conserved Lysine residue in the active center?
Do you see any correlation between the diploid and polyploid bamboo species for the evolution of WNK family ?

---

## Round 0.2 · Minor Revisions

Thanks for the manuscript update. In the final submission please check formatting style as suggested by reviewer #3. For example, gene names should be written in Italic font in the text.

In addition, please address the following items noted by a Section Editor:

1. It would be really helpful if the authors could add an extra sentence or two at the end of their introduction to better state why identifying WNK genes in bamboo is important or necessary. At the moment they basically just say 'we need to identify them because no one has before'. However, that can't be the only reason. In the discussion, the authors mention GMO type applications and the fact that bamboo has variable yield in winter. Surely these could be added to the introduction to strengthen the rationale for identifying WNK.

2. Figure 1 caption doesn't explain what the numbers are above the branches - presumably these are the bootstrap support values.

3. Figure 5 and 7 captions don't explain what the 'whiskers' or error bars are - are they SD, SE or CI?

4. Figure 6 and 8 colour scheme may need to be revised - red/green combinations are not the best for figures as many readers may be red/green colour blind.

5. Consistency in Moso bamboo vs moso baboo vs P. edulis should be addressed throughout the manuscript and the figures/tables.

6. It's still not 100% clear why the authors focus so much on moso bamboo for reporting results when they actually looked at five different bamboo species. The reviewers commented on this too so I think the authors need to be very clear about why this is the focus.

Reviewer 1 ·

Basic reporting

No comment

Experimental design

No comment

Validity of the findings

No comment

·

Basic reporting

The paper entitled “Genome-wide identification and evolution of WNK kinases in Bambusoideae and transcriptional profiling during abiotic stress in Moso bamboo”by Liu identified 41 WNK genes in five bamboo species and analyzed the gene evolution, phylogenetic relationship, physical, cis-elements, conserved motifs, various abiotic stresses, PeWNK genes expression levels. In this regard, the new reported results would be very useful in future functional genomic studies of Bambusoideae WNK genes.The work is suitable for the journal and the results are interesting.

Experimental design

The methodology part is much better than before and provide the bioinformatics part along with the parameters used. Also, check basic rules for botanical publishing. The genus and species names should be written with italics in the text.

Validity of the findings

Here, the results were presented again with a clear connection with the previous studies and the importance of the present study. The conclusions section, as written, state the significance of results and what perspectives for the field given the results.

Additional comments

There are several ways how this paper can be improved.
For example,the genes names should be written with italics in the text.
Reference list should be rechecked and corrected according to the journal style. like "Transcriptome data of 26 different tissues of moso bamboo were obtained from the NCBI Short Read Archive database (SRX2408703) [19] "
The tile "PeWNK genes response to circadian rhythms" should be capitalized in some words.

---

## Round 0.3 · accepted · Accept

Thank you for the manuscript update. The reviewers had no critical remarks. All the technical remarks were taken into account. I endorse the publication.